# Three years of measurements of light-absorbing aerosols over coastal Namibia: seasonality, origin, and transport

Paola Formenti[1,$], Stuart John Piketh[2], Andreas Namwoonde[3], Danitza Klopper[2], Roelof Burger[2], Mathieu Cazaunau[1], Anaïs Feron[1], Cécile Gaimoz[1], Stephen Broccardo[2], Nicola Walton[2], Karine Desboeufs[1], Guillaume Siour[1], Matheus Hanghome[3], Samuel Mafwila[3], Edosa Omoregie[3], Wolfgang Junkermann[4], and Willy Maenhaut[5]

[1] LISA, UMR CNRS 7583, Université Paris Est Créteil et Université Paris Diderot, Institut Pierre Simon Laplace, Créteil, France

[2] School of Geo- and Spatial Science, Unit for Environmental Sciences and Management, North-West University, Potchefstroom, South Africa

[3] Sam Nujoma Marine and Coastal Resources Research Centre (SANUMARC), University of Namibia, Sam Nujoma Campus, Henties Bay, Namibia

[4] Karlsruhe Institute of Technology, Institute of Meteorology and Climate Research, IMK-IFU, Garmisch-Partenkirchen, Germany

[5] Ghent University (UGent), Department of Chemistry, Gent, Belgium

[$] corresponding author (paola.formenti@lisa.u-pec.fr)

**Abstract**

Continuous measurements between July 2012 and December 2015 at the Henties Bay Aerosol Observatory (HBAO; 22°S, 14°05'E), Namibia, show that, during the austral wintertime, transport of light-absorbing black carbon aerosols occurs at low-level into the marine boundary layer. The average of daily concentrations of equivalent black carbon (eBC) over the whole sampling period is 53 (± 55) ng m$^{-3}$. Peak values above 200 ng m$^{-3}$ and up to 800 ng m$^{-3}$ occur seasonally from May to August, ahead of the dry season peak of biomass burning in southern Africa (August to October). Analysis of three-day air mass back-trajectories show that air masses from the south Atlantic ocean south of Henties Bay are generally cleaner than air having originated over the ocean north of Henties Bay, influenced by the outflow of the major biomass burning plume, and from the continent, where the wildfires occur. Additional episodic peak concentrations, even for oceanic transport, indicate that pollution from distant sources in South Africa and maritime traffic along the Atlantic ship tracks could be important. While we expect the direct radiative effect to be negligible, the indirect effect on the microphysical properties of the stratocumulus clouds and the deposition to the ocean could be significant and deserve further investigation, specifically ahead of the dry season.

**1. Introduction**

Aerosol particles of natural and anthropogenic origin affect the Earth's climate and modulate the greenhouse effect of long-lived gases (Boucher et al., 2013). The extent of this modulation depends on their nature, in particular on their chemical composition and size distribution determining their interactions with radiation and clouds. Current understanding suggests that atmospheric aerosols increase the global outgoing shortwave radiation, enhancing the atmospheric albedo, thereby counteracting the warming effect of greenhouse gases (Boucher et al., 2013). However, light-absorbing aerosols, such as black carbon (BC) from fossil fuel combustion and biomass burning, can reduce the amount of outgoing radiation at the top of

atmosphere (TOA), finally adding to the greenhouse effect (Haywood and Shine, 1995;
Jacobson, 2001; Chung and Seinfeld, 2002; Bond and Bergstrom, 2006; Koch and Del Genio,
2010; Bond et al., 2013). The heating radiative effect of black carbon aerosols is either enhanced
or suppressed if they are above or below clouds, respectively (Keil and Haywood, 2003; Koch
and Del Genio, 2010). The local heating induced by light-absorption below clouds could modify
the cloud properties by enhancing the vertical motion and increasing the cloud cover and liquid
water content (Koch and Del Genio, 2010). Finally, by entrainment into clouds, BC-containing
aerosols could cause the cloud to evaporate and rise (Hansen et al., 1997) and reduce the cloud
mean drop size diameters, increase droplet concentrations and henceforth reflectivity (Seinfeld
and Pandis, 1997).
These processes are relevant to the western coast of southern Africa, pointed out by the latest
Intergovernmental Panel for Climate Change (IPCC) report as a region where future warming
and reductions in precipitation should be severe (Maure et al., 2018).
The west coast of southern Africa is characterised by a persistent and extended stratocumulus
cloud deck topping a shallow, stable marine boundary layer maintained by the cold sea-surface
temperatures of the Benguela Current (Cook et al., 2004; Tyson and Preston-Whyte, 2002), and
by high loading of light-absorbing aerosols, mostly from seasonal biomass burning in the austral
dry season (Swap et al., 2002), but possibly from various local and distant anthropogenic
activities including ship traffic and energy production (Piketh et al., 1999; Formenti et al., 1999;
Tournadre, 2004). Stratocumulus clouds are highly reflective and efficient in modifying the net
radiative balance at TOA (Boucher et al., 2013). However, the mechanisms by which they could
interact with light-absorbing aerosols, and the direct and indirect effects of those interactions
on the regional radiative budget are largely unknown (Keil and Haywood, 2003; Flato et al.,
2013; Myhre et al., 2013; Xu et al., 2014).
To address these questions, a large observational effort was initiated in the last few years by a
number of coordinated intensive airborne and ground-based field campaigns, analysis of
spaceborne observations, and climate modelling (Zuidema et al., 2016). These experiments
focused on the dry season period between July and October, when biomass burning aerosols
contribute by optically-dense plumes with instantaneous aerosol optical depth (AOD)
systematically larger than 0.5 at mid-visible wavelengths (Swap et al., 200é). The emission,
transport and direct radiative effect of light-absorbing carbonaceous aerosols by biomass
burning aerosols also motivated previous experiments, such as the Southern African Regional
Science Initiative (SAFARI 2000, Swap et al., 200é) and the Southern African Fire-Atmosphere
Research Initiative (SAFARI, Andreae et al., 1996).
However, little is known about the aerosol concentrations and properties outside this season.
To fill this gap, this paper present the first results of the mass concentrations of light-absorbing
carbonaceous aerosols on the Atlantic coast of Namibia from three years of observations at the
Henties Bay Aerosol Observatory (HBAO, 22°S, 14°05'E) long-term ground-based surface
station.
Measurements of the mass concentrations of equivalent black carbon (eBC) recorded between
July 2012 and December 2015 in the marine boundary layer below the stratocumulus deck are
analysed to gather new knowledge on importance and seasonality. Observations are coupled
with calculations of air mass back-trajectories to identify the dominant transport patterns and
quantify their contributions. A comparison to the MERRA-2 model reanalysis is performed.
**2. Methods**
Surface observations of aerosol particles are conducted at the Henties Bay Aerosol Observatory
(HBAO, www.hbao.cnrs.fr), a recent regional station in the Global Atmosphere Watch (GAW)
Programme of the World Meteorological Organization (WMO). The research centre is located

on the Sam Nujoma Marine and Coastal Resources Research Centre (SANUMARC) of the University of Namibia in Henties Bay (22°S, 14°05'E), Namibia (Figure 1). Henties Bay is a small town in an arid environment with no vegetation, no industrial activity and very little traffic. Energy usage is predominantly a mix of electricity and gas, with some solid fuel combustion (wood) due to low availability (A. Namwoonde, 2017, *pers. comm.*). The monitoring site, situated on the University campus, is located on the coast approximately 100 m from the shore line. To the east are the Namibian Gravel Plains, at 3 km to the south of the campus is the town of Henties Bay, and to the north is the Omaruru Riverbed (river mouth approximately 100 m from SANUMARC). The population of Henties Bay ranges between 4 600 and 6 000 inhabitants, according to the Namibia 2011 population and housing census (main report available at http://cms.my.na/assets/documents/p19dmn58guram30ttun89rdrp1.pdf).

## 2.1. Measurements of light-optical attenuation

Instruments at HBAO operate from a roof terrace at approximately 30 m above the ground. The terrace hosts the sampling inlets, from which air is drawn into a laboratory room located underneath by straight stainless-steel pipes to avoid particle losses. The optical attenuation of light (ATN) by aerosol particles smaller than 1 μm in aerodynamic diameter was measured by a single-wavelength aethalometer (model AE-14U, Magee Sci., Berkeley, CA) operating at 880 nm and sampling at 3.5 ($\pm$ 0.1) L min$^{-1}$ from a certified $PM_1$ inlet (BGI Inc., Waltham, MA). The physical principle of operation of the aethalometer is detailed in Hansen et al. (1984). Measurements were performed at a 5-min time resolution and stored on a data logger (model CR-1000, Campbell Sci. Ltd.). The original data set was screened to eliminate spikes and peaks lasting less than two hours, generally associated with open fires for barbequing meat. The data record extended from July 2012 to December 2015, with an extended data gap between January and July 2014 due instrument maintenance.

The Lambert-Beer law relates the temporal variation of the measured light-attenuation (ATN)
due to aerosol particles collected on a quartz fibre tape to the mass concentration of eBC (in μg
m$^{-3}$). This is based on the fact that black carbon is the strongest light-absorber in the near
infrared (Kirchstetter et al., 2004; Caponi et al., 2017).
The operational equation linking eBC to the attenuation (ATN) measured by the aethalometer
is

$$eBC = \frac{1}{MAC_{BC}} \left( \frac{1}{C \cdot R(ATN)} \right) \left( \frac{A}{V} \frac{\Delta ATN}{\Delta t} \right) \qquad (1)$$

where $A$ represents the area of the aerosol deposit on the filter, $V$ the volumetric flow rate, and
$\Delta ATN/\Delta t$ is the variation rate of attenuation with time. The terms C and R(ATN) account for
measurement artefacts that artificially increase absorption estimated from attenuation
measurements. The term C takes into account the multiple scattering effects on the filter due to
both the filter fibers and the aerosol particles embedded in it. The factor R(ATN) accounts for
the shadowing effect occurring with time as high concentrations of absorbing particles are
collected on the filter. Published values of the C parameter at 660 and 880 nm range between
1.75 and 6.3 depending on the nature of the light-absorbing aerosols, the measurement
environment, and finally on the parametrisation of the corrections (Weingartner et al., 2003;
Arnott et al., 2005; Schmid et al., 2006; Collaud-Coen et al., 2010; Segura et al., 2014; Saturno
et al., 2017; Di Biagio et al., 2018). These authors show that, regardless of location, values
below 3.5 are appropriate for moderately absorbing aerosols whose single scattering albedo
(ω$_0$) is above 0.8 at 660 nm. For Mace Head, a coastal site with prevailing marine North Atlantic
air masses, Collaud-Coen et al. (2010) reported a mean C value of 3.44 (± 0.21), which we used
for HBAO, neglecting any possible wavelength dependence. The parametrisation of the
shadowing effect R(ATN) depends on $\omega_0$, henceforth, on the availability of concurrent
measurements of the scattering coefficient. This is the case at HBAO, where, however,
scattering is measured on the $PM_{10}$ and not on the $PM_1$ fraction as attenuation is, preventing a
meaningful estimate of $\omega_0$. In this case, Collaud-Coen et al. (2010) recommended the
Weingartner et al. (2003) correction, leading to a mean value of the R parameter of 0.93, which
we assumed for the further analysis.
The other crucial parameter in Equation (1) is the mass absorption efficiency of eBC ($MAC_{BC}$,
units of $m^2\ g^{-1}$). Many authors have reported values in the range 5-20 $m^2\ g^{-1}$ at wavelengths
between 550 and 870 nm, and related this variability to the chemical state and age of black
carbon aerosols (Liousse et al., 1993; Petzold et al., 1997; Martins et al., 1998; Kirchstetter et
al., 2003; Hansen, 2005; Bond and Bergstrom, 2006; Knox et al., 2009; Subramanian et al.,
2010; Bond et al., 2013; Zanatta et al., 2016). More recently, Zuidema et al. (2018) reported
that the $MAC_{BC}$ at 648 nm at Ascension Island, farther west than HBAO, and at times in its
outflow, varied between 14.1 $m^2\ g^{-1}$ in June to 10.7 $m^2\ g^{-1}$ in July to October. When
extrapolated, these values result in a $MAC_{BC}$ at 870 nm between 7.9 and 10 $m^2\ g^{-1}$. Their
average and standard deviation (9.0 ± 1.5 $m^2\ g^{-1}$) was retained in our analysis.
**2.2. Supporting data**
In 2013, the mass concentration of particles of diameter smaller than 2.5 μm in equivalent
aerodynamic diameter ($PM_{2.5}$) was sampled by a Tapering Element Oscillating Microbalance
(TEOM, model 1400a, Rupprecht and Patashnick, Albany, New York, USA) operating from a
certified $PM_{2.5}$ inlet (also from Rupprecht and Patashnick). The total flow rate at the inlet was
16.7 L $min^{-1}$ to ensure the correct functioning of the inlet, and the sampling flow rate driving
the aerosol-laden air to the microbalance was 3 L $min^{-1}$. The temperature of the sample stream
was kept constant at 50ºC.
Three-dimensional air mass back-trajectories are calculated using the NOAA HYbrid Single-
Particle Lagrangian Integrated Trajectory Model (HYSPLIT; Draxler and Rolph, 2015). The
model uses the $1° \times 1°$ latitude-longitude grid, reanalysis meteorological database. The 6-hourly
reanalysis archive data are generated by the NCEP's GDAS (NCEP: National Centers for
Environmental Prediction; GDAS: Global Data Assimilation System) wind field reanalysis.
Further information can be found at https://rda.ucar.edu/datasets/ds083.2/.
ERA-Interim reanalysis data (Dee, et al. 2011) from the European Center for Medium Range
Weather Forecast (ECMWF) are used in the analysis of synoptic scale circulation patterns
associated with the identified dominant air mass transport to Henties Bay. The 6 hourly (0, 6,
12, 18 UTC) analysis data (0.75 x 0.75 degrees) at mean sea level pressure (MSLP - variable
151.128) and the 500 hPa geopotential height (ZG500 -- variable 129.128) are used for this
study. Data sets were normalised (MSLP / 100 and ZG500 /100) using Climate Data Operators
(CDO) (Schulzweida et al., 2006) and plotted with 2 hPa and 10 hPa isobaric intervals for
MSLP and 500 hPa level, respectively.
Surface black carbon concentrations predicted by the Modern-Era Retrospective analysis for
Research and Applications, Version 2 (MERRA-2, Gelaro et al., 2017), sampled at HBAO and
at a number of other sites (Zuidema et al., 2016) are used for comparison.
The HTAP_V2 dataset is used for gridded emission of anthropogenic black carbon for the year
2010 (Janssens-Maenhout et al., 2015). It consists of 0.1° x 0.1° grid-maps. HTAP_V2 uses
nationally reported emissions combined with regional scientific inventories in the format of
sector-specific grid maps. The grid maps are complemented with EDGARv4.3 data for those
regions where data are absent. Anthropogenic activities producing black carbon aerosols
comprise aviation, transportation, energy production, industries, ship traffic, residential and
agricultural burning.

## 3. Results

### 3.1. Temporal variability of eBC concentrations

Figure 2 shows daily and monthly averages of the eBC concentrations measured at HBAO between July 2012 and December 2015. Daily averages excluded spikes and peak values occurring on short time scales, less than 1-2 hours, resulting from contamination of local activities (episodic traffic and occasional open fires for barbequing meat).

The daily mean average of 53 ($\pm$ 55) ng m$^{-3}$ is in accordance with previous observations in remote locations of the world shown in Table 1 (Bodhaine, 1995; Andreae et al., 1995; Derwent et al., 2001; von Schneidemesser et al., 2009; Marinoni et al., 2010; Sheridan et al., 2016). Andreae et al. (1995) found eBC mass concentrations lower than 50 ng m$^{-3}$ along a cruise transect at 19°S over the south east Atlantic between Brazil and Angola, except when approaching the African continent, when concentrations increased in the range 50-150 ng m$^{-3}$, indicating a strong continental influence in this otherwise pristine environment. Additional published research, also in Table 1, reports absorption coefficients that would lead to comparable eBC concentrations (Bodhaine, 1995; Clarke, 1989; Quinn et al., 1998). For contrast, eBC mass concentrations in lofted layers above the marine boundary layer in the range 0.1-6 µg m$^{-3}$ were reported for aged biomass burning haze (Kirchstetter et al., 2003; Formenti et al., 2003; Eatough et al; 2003), and up to 5-40 µg m$^{-3}$ for fresh biomass smoke plumes (Kirchstetter et al., 2003).

Figure 2 also shows an apparent seasonal variability in eBC, further highlighted by the monthly means and by the comparison with PM$_{2.5}$ mass concentration measurements performed at the site during 2013, which conversely, did not display any particular seasonal cycle (Figure S1), likely because dominated by sea salt. Concentrations increase in the austral winter from May to July, and decrease from August to April. The increase from May to July is well captured by the MERRA-2 reanalysis (also shown in Figure 2), according to which, however,

concentrations only start decreasing after September. The observed seasonality is somewhat
surprising in that it precedes the seasonal maximum of the biomass burning fire season in
southern Africa, peaking in the austral dry season from August to October (Swap et al., 2002).
As previously stated in this paragraph, data constituting the time series have been screened to
exclude short-term variability (less than one-two hour time intervals) to exclude isolated and
episodic sources. Peaks of eBC driving the seasonal increase in the May-to-August period are
long-lasting, extending between 6 and 11 hours, and occurring during both day- and night-time.
This suggests that transport is the cause of the seasonal peaks. The following section explores
this hypothesis and attempts the quantification of attribution of eBC peaks to specific transport
patterns.
**3.2. General atmospheric circulation driving air mass transport**
Transport to the west coast of Namibia is influenced by four general circulation patterns:
baroclinic westerly waves, barotropic easterly waves, the semi-permanent south Atlantic high
pressure and a continental high pressure circulation. The relative influence of each circulation
pattern is highly seasonal and driven by the meridional migration towards the north in austral
winter and to the south in summer. The origin of the air parcel over the southern ocean is linked
to the passage of a westerly wave and front that propagates towards the subcontinent form the
south west. These are Rossby waves that form as a result of the extratropical temperature
gradient with a maximum impact on the weather over southern Africa in winter. The easterly
waves are trade winds that are associated with the position of the Inter Tropical Convergence
Zone (ITCZ) that reaches a maximum over the subcontinent during summer (Tyson and
Preston-Whyte, 2014; Taljaard, 1994). The semi-permanent high pressure system (anticyclone)
results from the descending limb of the Hadley circulation that interacts with the
aforementioned waves. The south Atlantic high pressure system will ridges behind a passing
westerly wave in the direction of maximum cold air advection. Conversely, a strengthening
anticyclone will block propagations of these waves and induce a strong persistent continental
high pressure. Air masses reaching the sampling station have been found to originate from the
adjacent Atlantic Ocean and various locations over the continental subcontinent. Coastal lows
are induced along the west coast of Namibia and result in offshore flow ahead of westerly
waves. This low pressure system forms localised cyclonic circulation that includes onshore flow
in the north and offshore flow in the south of the low pressure cell (Tlhalerwa et al., 2005;
Tyson and Preston-Whyte, 2014).
**3.3. Identification of air mass transport pathways impacting HBAO**
Three-day back-trajectories calculated daily between July 2012 and December 2015 are
grouped according to the progression of the general synoptic circulation patterns and assigned
to 8 geographical sectors according to the position of their end point, shown in Figure 3. Four
sectors (G1 to G4) correspond to oceanic air masses and sectors (G5 to G7) to transport from
the continent. The last sector (G8) describes air masses recirculating around the sampling site
for most of the three day period. Figure 3 also shows the 2010 black carbon aerosol regional
emission HTAP_V2 inventory grid map from anthropogenic activities. Emissions are low in
Namibia and neighbouring countries as Botswana and the west-central South Africa. Areas of
higher emissions are Angola (with an hotspot in correspondence with the capital city Luanda),
costal South Africa, particularly to the east, but also to the south in the Cape Town greater area,
but mostly in the South African Highveld (27°S, 28°W) where the energy production is
concentrated. The open-ocean and coastal ship tracks are also evident.
The monthly distribution of fire counts from 2012 to 2015 provided by MODIS/Aqua is shown
in Figure 4. Although some interannual variability exists, the image record is consistent in
showing that the fire season in Southern African starts towards April and extends until October.
The major source areas are north of Namibia (Angola, Zambia), in South Africa (to the east and
along the south coast) and in Mozambique. In Namibia, fire counts are seen towards the north,
around the Etosha Pan desert.
The seasonal contribution of these air masses transport pathways is shown in Figure 5. Sectors
G1 to G4 represent the most common air flow pathway (73% out of 1279 calculated back-
trajectories). The Southern Atlantic Ocean transport (G1 and G2) is the dominant surface
circulation along the west coast, resulting from the northward moving limb of the surface south
Atlantic high pressure. Air masses originate over the southern Atlantic Ocean, as far south as
55°S (sector G2, representing approximately 66% of the air mass occurrences). During summer
this is predominantly a function of the most southerly location of the centre of the south Atlantic
high (Figure 6). In winter transport results from a complex interaction between the westerly
waves propagating from the south west over the subcontinent and the reestablishment of the
south Atlantic High in the westerly waves wake. Initially, air transport is towards the east and
is then directed northwards along the west coast to Namibia (Figure 7). The distance covered
by these air masses is several thousand kilometres due to the high wind speeds associated with
the initial transport in the cyclonic circulation.
Sectors G3 and G4 describe transport from the tropical regions of the Atlantic Ocean. The
onshore flow towards the sampling site forms as a westerly wave advances. A shallow, localised
cell of low pressure (cyclonic circulation) is induced along the west coast with a diameter of
approximately 200 km. Towards the north of the cell onshore flow occurs, while in the southern
portion of the cell the flow is offshore. This circulation has also been shown to induce the dust
plumes that blow off the Namibian coast over the Atlantic Ocean from ephemeral river beds
along the west coast of Namibia (Tlhalerwa et al., 2005). The low, referred to as a coastal low,
then propagates southwards behind the surface front. It is possible for air that is moved offshore
in the easterly wave over Angola to be caught up in this more near shore circulation.
Transport to HBAO from the continent (sectors G5-G7) occurred on 19% of the total days
(sectors G5, G6 and G7). These transport pathways are directly linked to the position of the
easterly wave over the subcontinent as well as the position of the trough line associated with
the wave. As pointed out earlier, the position of the easterly wave is highly seasonal. In general,
air is transported across the subcontinent and exits to the Atlantic Ocean in the westerly
transport. The low pressure trough moves across the subcontinent. The position of the trough
also determines the exact pathway of transport as well as the sector in which air masses
originate. If the trough is situated along the west coast it forms a west coast trough that
facilitates flow along and close to the west coast. During summer the easterly wave reaches to
the southern tip of southern Africa. This leads to transport of air from areas of South Africa,
including the highly industrialised Highveld region (Figure 8a). In winter the easterly wave
seldom reaches south of 25ºS. Air masses during this season are more likely to originate over
the central portion of southern Africa (Figure 8b).
Finally, sector G8 is associated with air masses originating within 100 km of HBAO (Figure
9), either from land or from the ocean, and representing about 8% of the air mass occurrences
(Figure 5). This circulation, only occurring in the second half of each year (Figure 5), is linked
to the formation of a low pressure heat cell close to the west coast of Namibia centred at about
the latitude of Henties Bay. Despite this being cyclonic flow the circulation is closed (Figure
9)and therefore represents transport from close to the sampling site. The heat low is always
embedded in an easterly wave or west coast trough. Centres of low pressure form along the
west coast producing local and mesoscale circulation from the interior of Namibia to the coast.
This flow pattern is distinguishable from a coastal low as it is centred on the subcontinent
whereas the coastal low is always centred on the coast just offshore.

**3.4. Contribution of air transport patterns to the measured eBC**

Figure 10 illustrates the contribution of the air mass sectors G1-G8 to the eBC mass concentrations measured at HBAO and those estimated by the MERRA-2 reanalysis. This has been done by calculating the distribution of eBC values per group.

Although the absolute values differ by a factor of 2-3, measurements and reanalysis show some consistent temporal features. Episodic high values of eBC concentrations occur independently on the origin of the air mass. The southern Atlantic oceanic air masses (sectors G1, G2 and G3) and the continental G7 sector, corresponding to the low population density semi-arid region of the Karoo, in South Africa, display the lowest concentrations. In particular, the oceanic sectors are characterised by a south-to-north gradient, the highest mean concentrations being from sectors G4, offshore northern Namibia and Angola, comparable to those from the continental sectors G5 and G6, and G8, representing recirculating air masses. Measurements at HBAO indicate that the contributions of sectors G5 and G6 are equivalent, while sector G5 is the largest contributor according to the MERRA-2 reanalysis.

**4. Discussion and conclusions**

This papers present the first long-term time series of equivalent black carbon concentrations in the marine boundary layer on the south-east Atlantic coast offshore southern Africa. Observations were conducted at the Henties Bay Aerosol Observatory, in Namibia, between July 2012 and December 2015.

Higher concentrations of eBC on the western coast of southern Africa are observed from April to July within continental and marine air masses north of 30°S (sectors G4, G5 and G6). Daily eBC peak concentrations at HBAO do not exceed 800 ng m$^{-3}$, and are seldom larger than 200 ng m$^{-3}$, lower than measured at Ascension Island, approximately 1500 kilometres downwind of coastal Namibia and located along the main outflow pathway from southern Africa to the

Atlantic Ocean (Swap et al., 1996; 2002; Adebiyi and Zuidema, 2016; Zuidema et al., 2018).
The seasonality of the eBC concentrations observed at HBAO corresponds to the seasonal shift
from southern to northern circulation at the surface, and is in phase with the April onset of the
fire season in southern Africa (Figure 5). The seasonal increase at HBAO is also well captured
by the MERRA-2 reanalysis model, but it occurs earlier than reported by Zuidema et al. (2018)
at Ascension Island (June to August). This seems to indicate that HBAO is on a minor branch
of the transport pathway of the continental biomass burning smoke plume from continental
southern Africa compared with the biomass burning plumes that reach Ascension Island. The
MERRA-2 reanalysis shows higher concentrations than measured at HBAO and suggests that
the period of high concentrations should persist until September rather than August as in the
HBAO measurements. This points out to the inherent degree of uncertainty in our estimates.
The correction factors (filter loading and multiple scattering corrections) needed to convert the
measured attenuation into a value eBC concentration are assumed and not evaluated from
concurrent measurements, and set to fixed values as the aerosol at HBAO would derive from a
single source type. We do not deal either with potential changes of the aerosol properties due
to ageing. Differences could also be due to the representation of the timing and extent of a
southward shift of the easterly during summer. Although these issues cannot be resolved with
the present dataset, they question the representation of the transport of smoke plumes at the
subcontinental scale of southern Africa.
There is no doubt that the transport of wildfire smoke is the major regional source of the eBC
aerosols for the western coast of Namibia. However, the presence of episodic outliers and the
relatively elevated concentrations observed for oceanic air masses originating south of HBAO
(sectors G1 to G3 in Figure 7) suggests that additional sources could contribute to the load of
light-absorbing aerosols in the marine boundary layer. In particular, the contribution of the
coastal and open ocean maritime shipping routes in the south Atlantic ocean (Tournadre, 2014;
Fraser et al., 2016; Johannson et al., 2017), and that of long-range continental anti-cyclonic
transport from the industrial areas of the South African Highveld, showing up in Figure 3
(Piketh et al., 2002), should be further explored.
By the very rough assumption of the mass fraction of black carbon to the total fine aerosol
(10%, Bond et al., 2013), we estimate that the mean fine mass of aerosols containing eBC would
be 0.5 ($\pm$ 0.5) $\mu$g m$^{-3}$. For comparison, the mean PM$_{2.5}$ mass concentration at HBAO was 14 ($\pm$
11) $\mu$g m$^{-3}$ in 2013 (Figure S1).
These aerosols below clouds would have a negligible direct radiative effect. There are almost
no AERONET measurements of the aerosol optical depth (AOD) at HBAO concurrent to the
eBC data series. However, Figure S2 shows the time series of the AERONET level 2.0 AOD
of the fine and coarse mode aerosols (AOD$_F$ and AOD$_C$) evaluated by the O'Neill et al. (2003)
algorithm between December 2011 and May 2012, and then from May to December 2015.
Figure S2 shows that the AOD$_F$ varies significantly from background values in the December
2011-May 2012 period (average 0.05 $\pm$ 0.03) to peak values of 0.4 and higher during August,
September and October 2015, when the transport of biomass burning occurs in the free
troposphere (Swap et al., 2003). The AOD$_C$, contributed essentially by sea salt, is relatively
invariant with time. There is no process other than biomass burning that would inject aerosols
above the marine boundary layer, henceforth we can consider the mean value for December
2011-May 2012 as a reasonable evaluation of the optical depth of the fine mode of aerosols
below clouds, including eBC.
The eBC aerosols might act on the microphysical properties of the local stratocumulus clouds.
At Ascension Island, Zuidema et al. (2018) demonstrated the good correlation between the
concentrations of refractory black carbon and cloud condensation nuclei (CCN) at
supersaturations exceeding 0.2%. A similar effect should be expected at HBAO and could be
important, outside but also during the biomass burning season as the entrainment of biomass
burning aerosols from the free troposphere maybe be inhibited by the thermal inversions clear
air slots separating the elevated plumes and the marine boundary layer (Keil and Haywood,
2003; Haywood et al., 2003; Hobbs et al., 2003). Finally, by deposition, these low-level aerosols
could act on the biological activity for the oligotrophic south Atlantic gyre in summer,
providing with soluble nutrient species, such as dissolved nitrogen, phosphorous, and iron
(Guieu et al., 2005; Luo et al., 2008; Paris et al., 2010).
In conclusion, the chemical apportionment of the $AOD_F$ below cloud and the hygroscopic
properties of the eBC aerosols at HBAO deserves exploration by future refined experiments.
**Data availability**
Original data for measured light-attenuation at HBAO are distributed by the French national
AERIS data center (https://www.aeris-data.fr/direct-access-icare-2/). Treated data can be
obtained by email request to the first author of this paper.
**Author contributions**
Paola Formenti, Stuart John Piketh, Andreas Namwoonde, Samuel Mafwila, Edosa Omoregie,
Wolfgang Yunkermann and Willy Maenhaut designed the experiments and the sampling site.
Paola Formenti, Stuart John Piketh, Andreas Namwoonde, Mathieu Cazaunau, Anaïs Feron,
Cécile Gaimoz, Stephen Broccardo, Nicola Walton, Karine Desboeufs, and Mattheus
Hanghome performed the experiments.
Paola Formenti and Stuart John Piketh performed the full data analysis with contributions by
Danitza Klopper, Guillaume Siour, and Roelof Burger.
Paola Formenti, Stuart Piketh, Danitza Klopper and Roelof Burger wrote the paper with
comments from all co-authors.
**Competing interests**
The authors declare that they have no conflict of interest.

**Acknowledgments**


This work receives funding by the French Centre National de la Recherche Scientifique (CNRS)
and the South African National Research Foundation (NRF) through the "Groupement de
Recherche Internationale Atmospheric Research in southern Africa and the Indian Ocean"
(GDRI-ARSAIO) and the Projet International de Coopération Scientifique (PICS) "Long-term
observations of aerosol properties in Southern Africa" (contract n. 260888) as well as by the
Partenariats Hubert Curien (PHC) PROTEA of the French Ministry of Foreigns Affairs and
International Development (contract numbers 33913SF and 38255ZE).
We acknowledge the use of the HYSPLIT model from the NOAA Air Resources Laboratory
(ARL), the use of FIRMS data and imagery from the Land Atmosphere Near-real time
Capability for EOS (LANCE) system operated by the NASA/GSFC/Earth Science Data and
Information System (ESDIS) with funding provided by NASA/HQ, the archiving and
distribution of the HTAPv2 gridmap by the Emissions of atmospheric Compounds and
Compilation of Ancillary Data (ECCAD) database. The global gridmaps are a joint effort from
US-EPA, the MICS-Asia group, EMEP/TNO, the REAS and the EDGAR group to serve in the
first place the scientific community for hemispheric transport of air pollution. The static version
is available on this EDGAR website, but also the GEIA data portal and the ECCAD server.
ECCAD is the GEIA Global Emission InitiAtive database (www.geiacenter.org) and is part of
AERIS, the French data service for Atmosphere (www.aeris-data.fr).
MERRA-2 data are available at MDISC, managed by the NASA Goddard Earth Sciences (GES)
Data and Information Services Center (DISC). Thanks are due to A. Da Silva (NASA/Goddard
Space Flight Center, Global Modeling and Assimilation Office, Greenbelt, MD, USA) for
making the extractions at HBAO available.

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

 **Table captions**

**Table 1.** Values of mass concentrations of equivalent black carbon (eBC) from measurements
published in the literature for remote regions worldwide. When available, the specific
attenuation used to convert the measured attenuation to eBC is also reported.
**Figure captions**
**Figure 1:** Geographical location of the Henties Bay Aerosol Observatory (HBAO).
**Figure 2.** Top panel (a): comparisons of the time series of daily eBC mass concentrations (ng
m$^{-3}$) measured at HBAO and predicted by the MERRA-2 reanalysis. The light grey boxes
indicate periods of increasing concentrations. The light blue boxes indicate periods of
decreasing concentrations. Bottom panel (b) box and whisker plot representation of the
respective monthly variability.
**Figure 3.** Geographical boundaries of the sectors used to classify the air mass back trajectories
superimposed to the emission grid-maps at 0.1° x 0.1° degrees of black carbon aerosols from
anthropogenic activities for the year 2010 provided by the HTAP_V2 inventory. Emissions are
expressed in Tons.
**Figure 4**. Fire counts per pixels from MODIS/Aqua provided by the NASA Fire Information
for Resources Management System (FIRMS). Colours range from yellow (1 fire count per
pixel) to red (+100 fire count per pixel). The underlying image is the corrected reflectance (true
colour) measured by MODIS/Aqua on the first day of each months.
**Figure 5**. Seasonal variation in the transport pathways of air masses reaching HBAO between
2012 and 2015.
**Figure 6**. Case study of mean sea level pressure over the sub-continent and adjacent South
Atlantic Ocean for 16-19 November 2014 illustrating the synoptic circulation that results in the
transport of air masses from sector G1.
**Figure 7**. Case study of mean sea level pressure over the sub-continent and adjacent South
Atlantic Ocean for 1-4 June 3013 illustrating the synoptic circulation that results in the
transport of air masses from sector G2.
**Figure 8**. Case study of mean sea level pressure over the sub-continent and adjacent south
Atlantic ocean for A) summer (10-13 December 2013) and B) winter (6-10 July 2013)
illustrating the synoptic circulation that results in the transport of air masses from sector G5-
G7.
**Figure 9**. Case study of mean sea level pressure over the sub-continent and adjacent South
Atlantic Ocean for 13-16 December 2012 illustrating the synoptic circulation that results in the
transport of air masses from sector G8.
**Figure 10**. Contribution of air mass sectors to the eBC concentrations at HBAO from (a) our
measurements and (b) MERRA-2 reanalysis.

**Table 1.** Values of mass concentrations of equivalent black carbon (eBC) from measurements
published in the literature for remote regions worldwide. When available, the specific
attenuation $\sigma_{BC}$ used to convert the measured attenuation to eBC is also reported.

| Location | eBC (ng m$^{-3}$) | $\sigma_{BC}$, 880 nm (m$^2$ g$^{-1}$) | Reference |
|---|---|---|---|
| Tropical South Atlantic off southern Africa, 19°S | 50-150 | 10 | Andreae et al. (1995) |
| Nepal Climate Observatory Pyramid, Himalaya | 160 ± 296 | 6.5[$]* | Marinoni et al. (2010) |
| Summit, Greenland | <340 | ----- | von Schneidemesser et al. (2009) |
| Mace-Head, Ireland | 47-74 | 11 ± 3 | Derwent et al. (2001) |
| South Pole, Antarctica | 0.01-50 | 19 | Bodhaine (1995) |

[$] At 635 nm – measurements were conducted with a Multi-Angle Absorption Photometer
(MAAP 5012, Thermo Electron Corporation).

**Figure 1:** Geographical location of the Henties Bay Aerosol Observatory (HBAO).

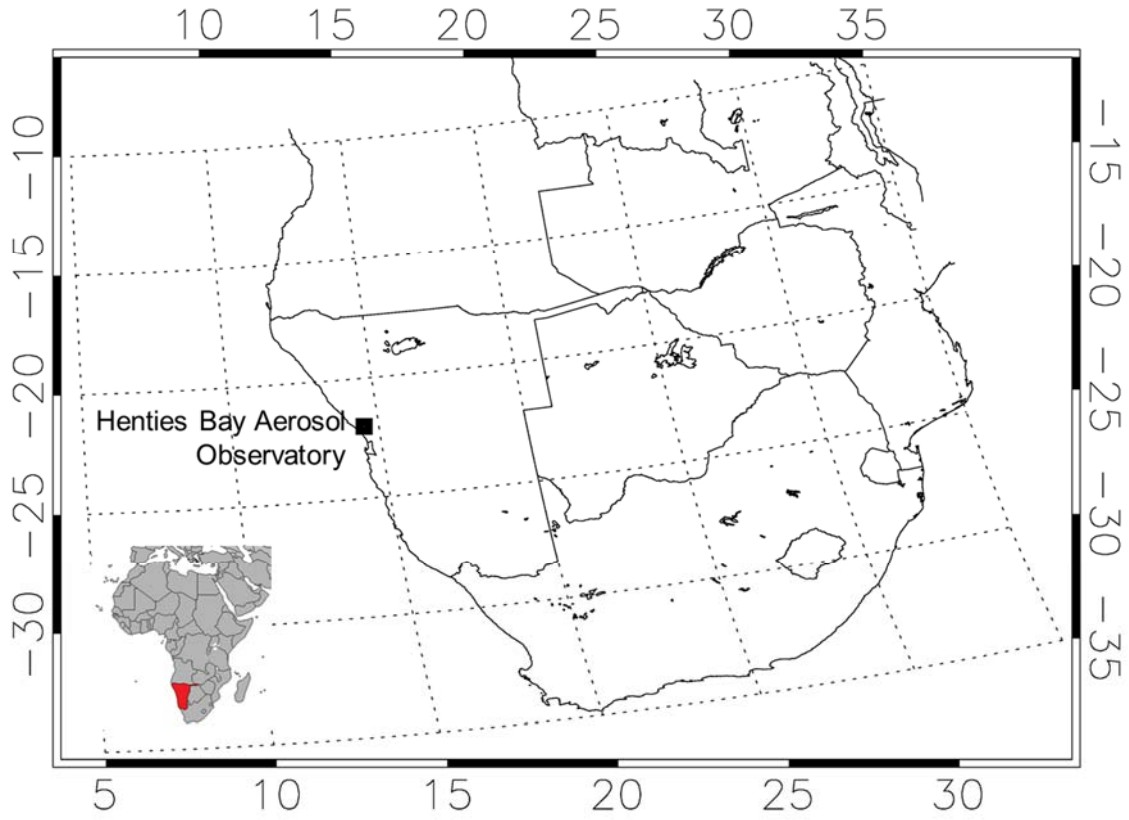



**Figure 2.** Top panel (a): comparisons of the time series of daily eBC mass concentrations (ng m-3) measured at HBAO and predicted by the MERRA-2 reanalysis. The light grey boxes indicate periods of increasing concentrations. The light blue boxes indicate periods of decreasing concentrations. Bottom panel (b) box and whisker plot representation of the respective monthly variability.

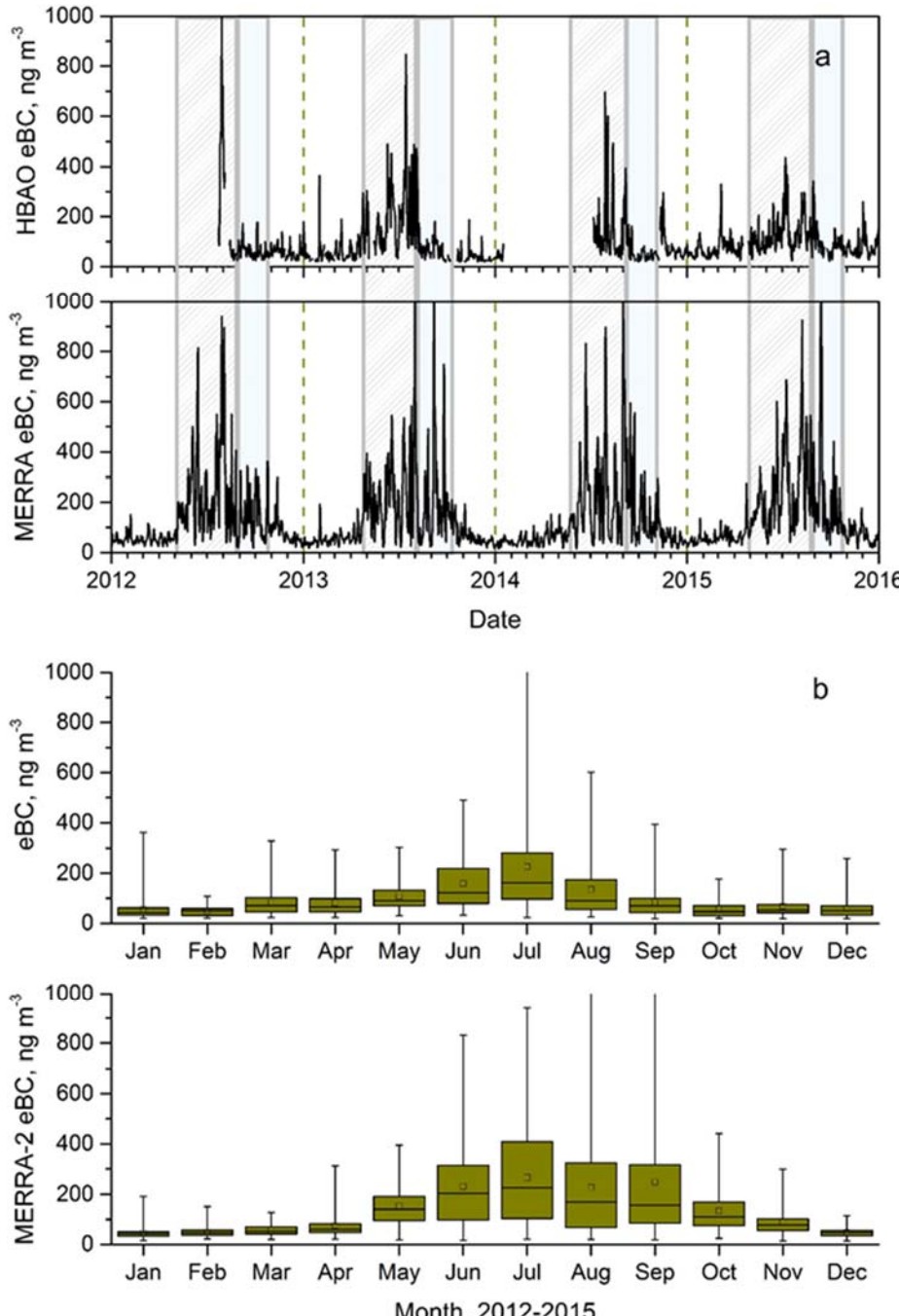

725

**Figure 3.** Geographical boundaries of the sectors used to classify the air mass back trajectories superimposed to the emission grid-maps at 0.1° x 0.1° degrees of black carbon aerosols from anthropogenic activities for the year 2010 provided by the HTAP_V2 inventory. Emissions are expressed in Tons.

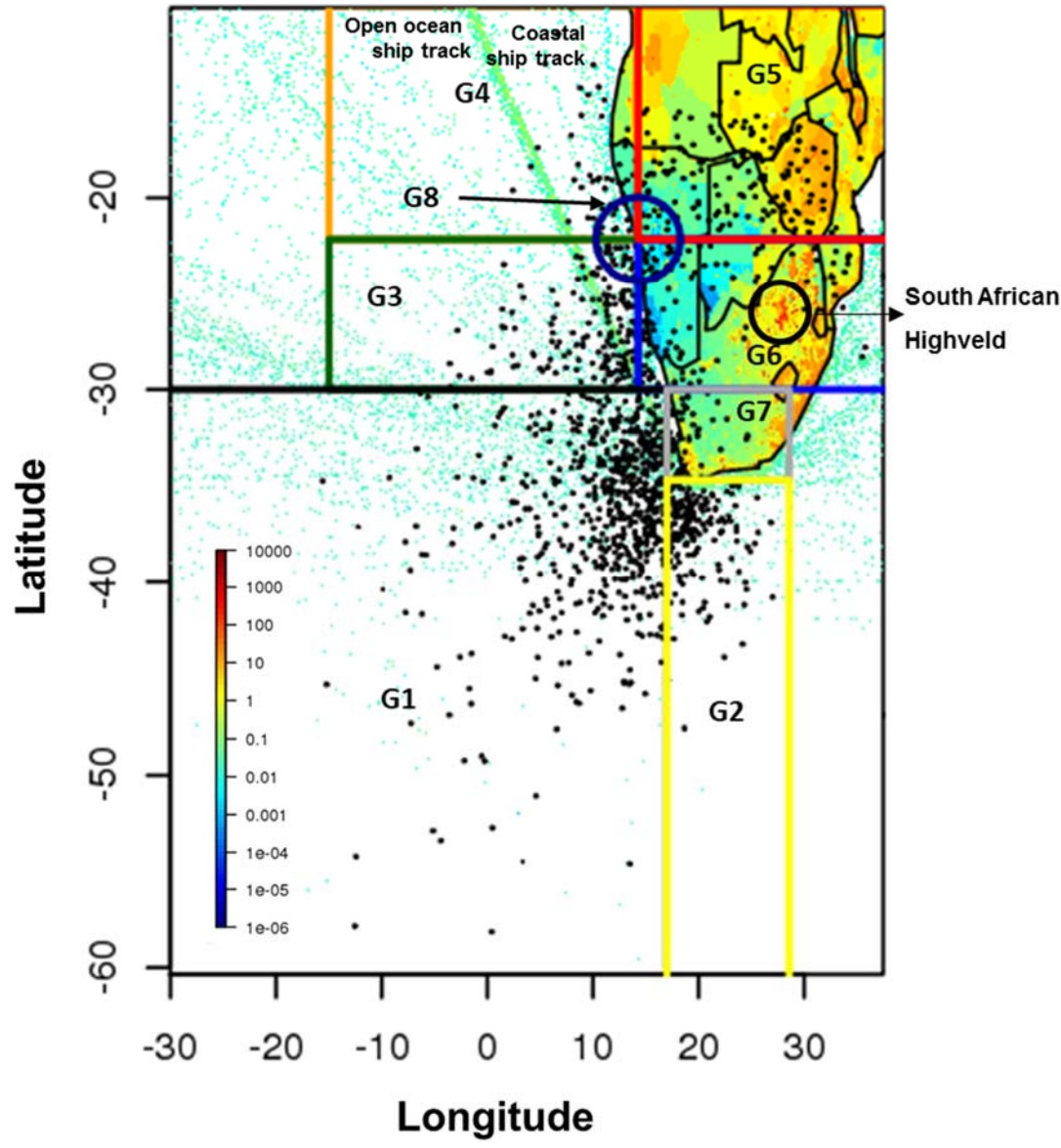

730

**Figure 4.** Fire counts per pixels from MODIS/Aqua provided by the NASA Fire Information for Resources Management System (FIRMS). Colours range from yellow (1 fire count per pixel) to red (+100 fire counts per pixel). The underlying image is the corrected reflectance (true colour) measured by MODIS/Aqua on the first day of each months. Sectors G1 to G8 are indicated over the first image

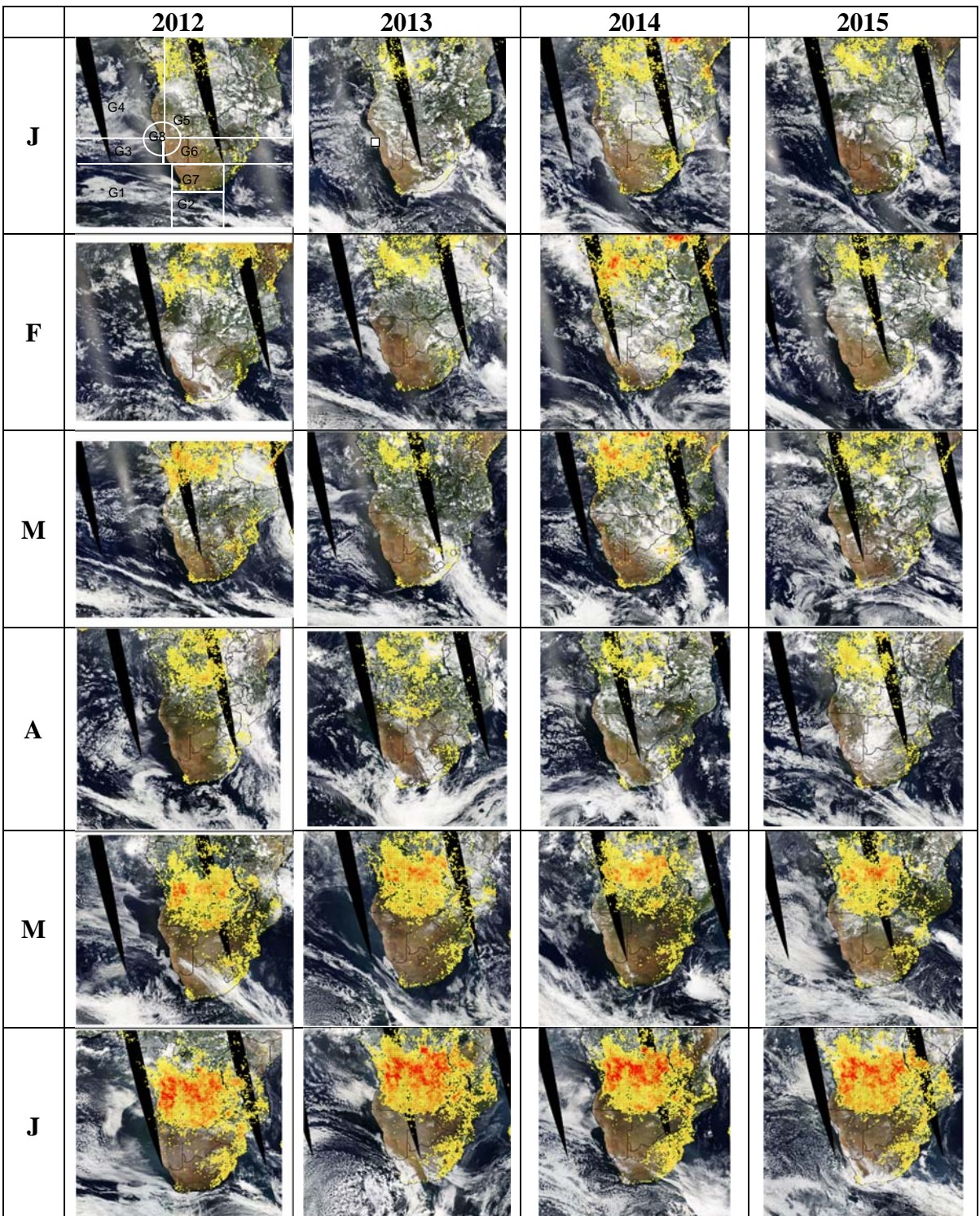

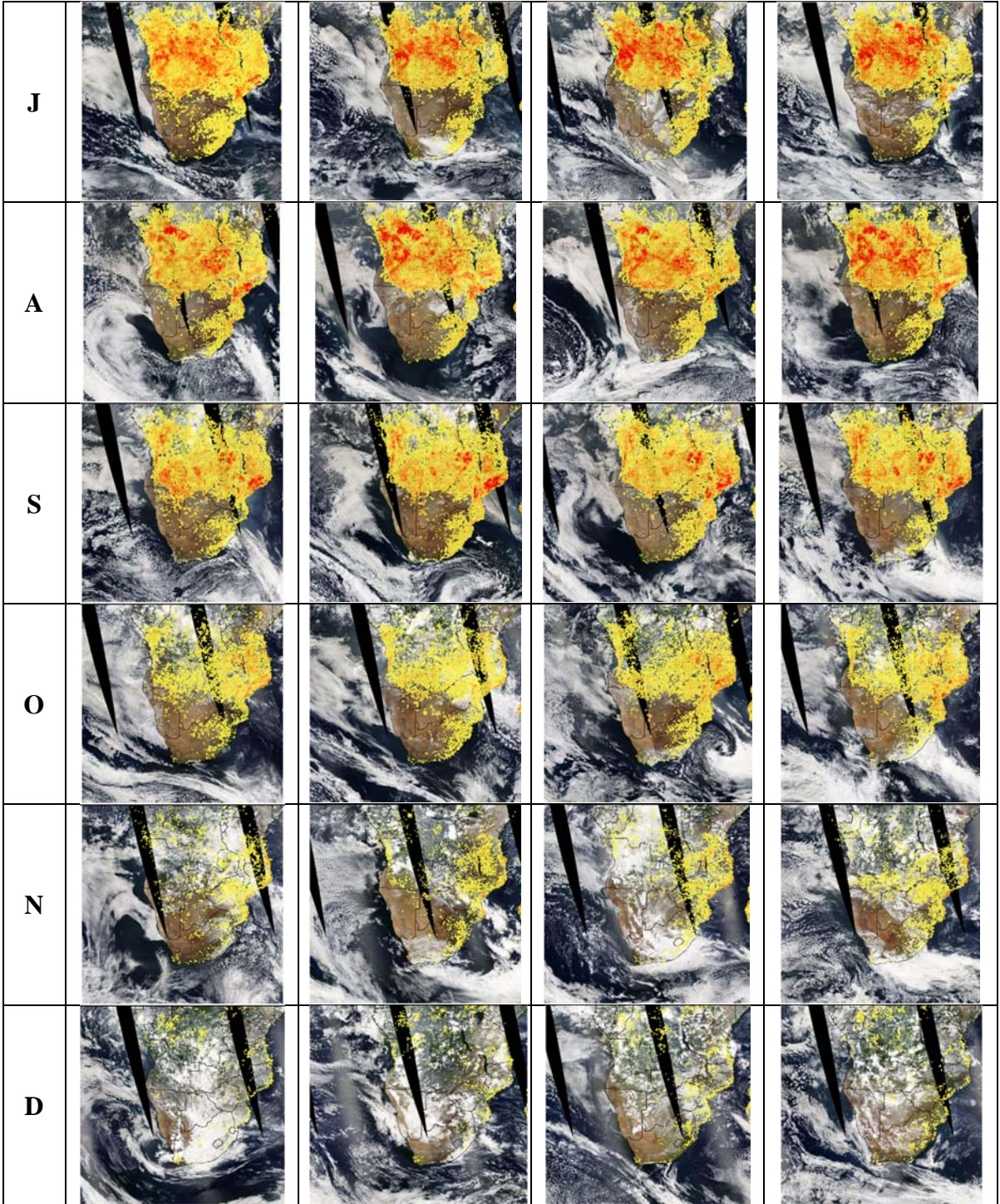


**Figure 5**. Seasonal variation in the transport pathways of air masses reaching HBAO between
2012 and 2015.

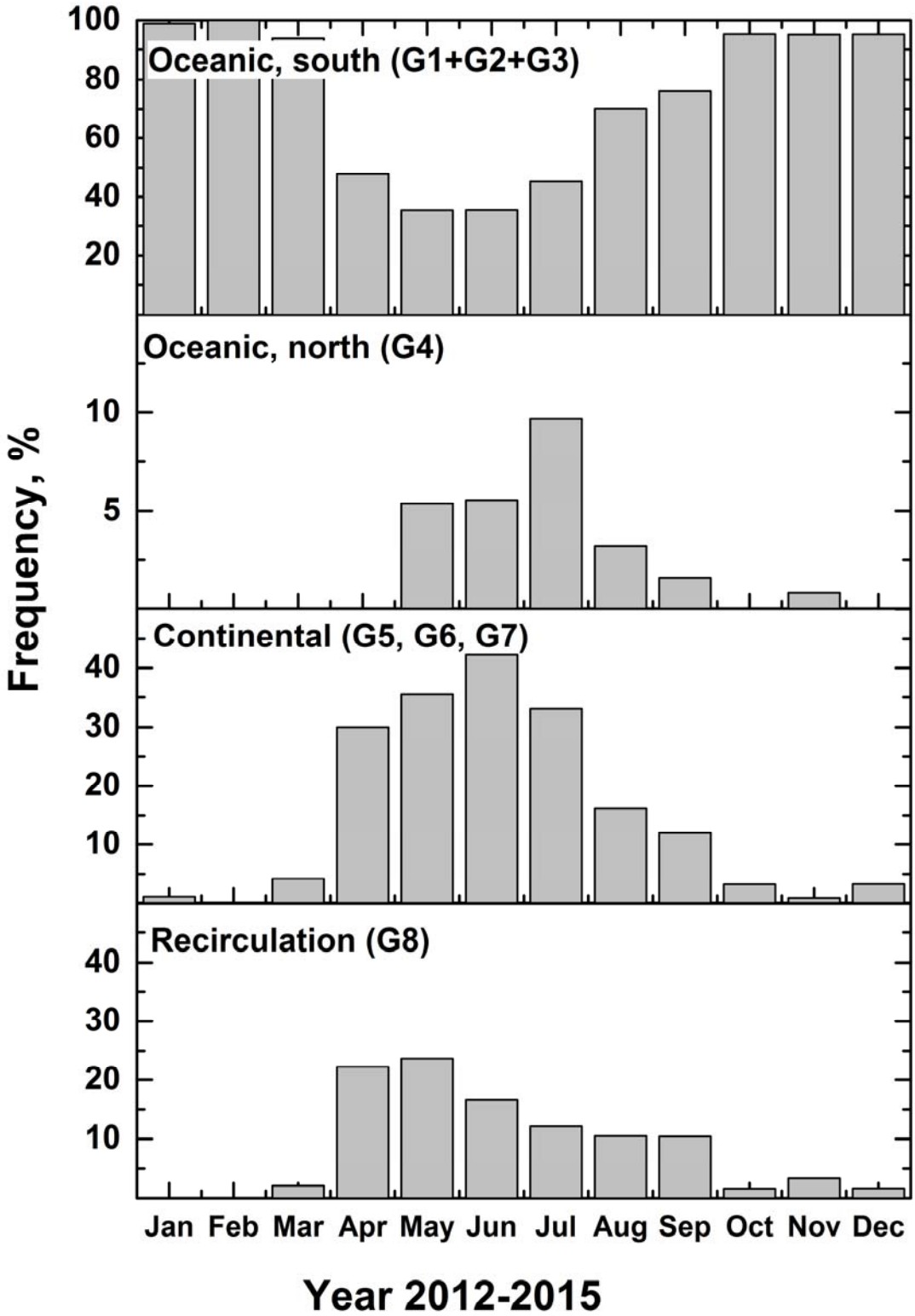



**Figure 6**. Case study of mean sea level pressure over the sub-continent and adjacent South
Atlantic Ocean for 16-19 November 2014 illustrating the synoptic circulation that results in the
transport of air masses from sector G1.

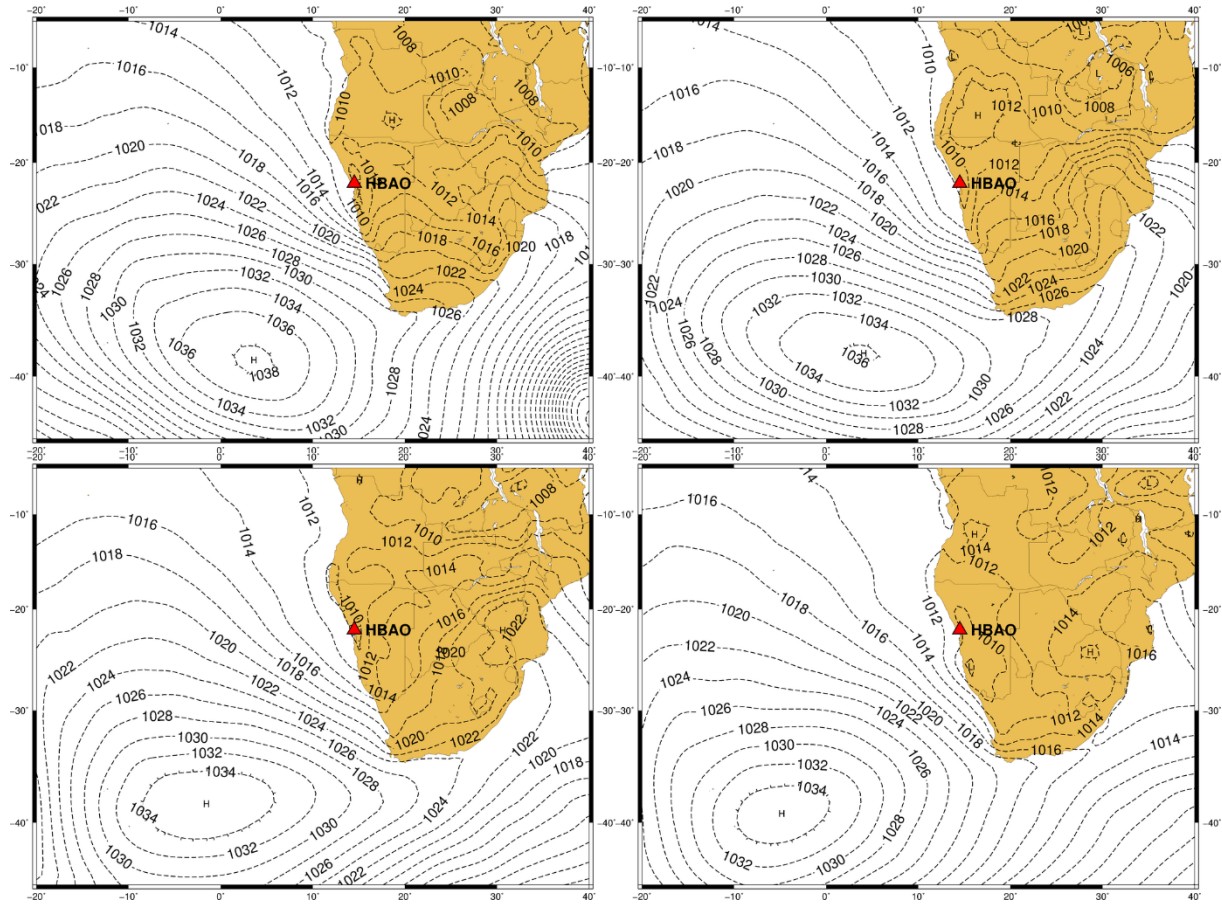





**Figure 7**. Case study of mean sea level pressure over the sub-continent and adjacent South
Atlantic Ocean for 1-4 June 3013 illustrating the synoptic circulation that results in the transport
of air masses from sector G2.

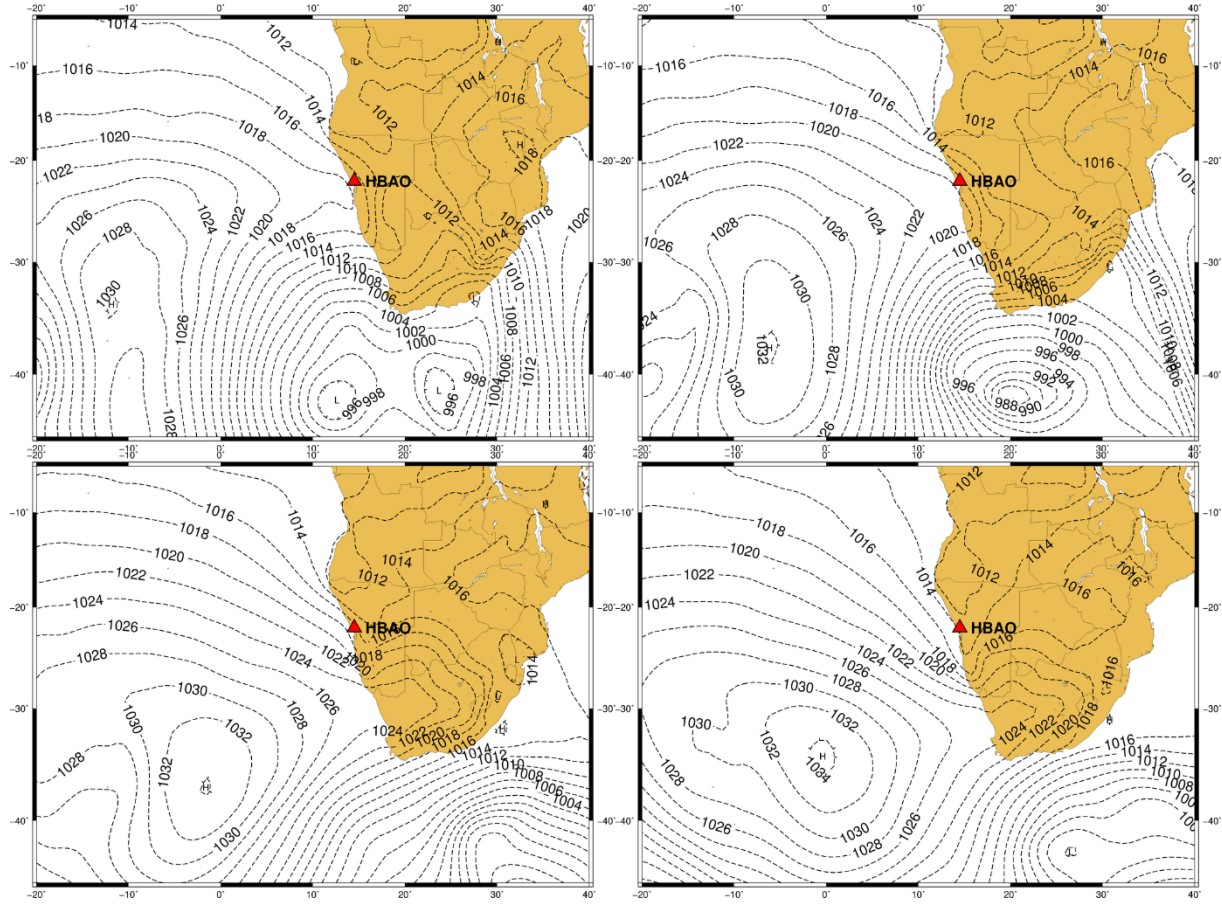



**Figure 8**. Case study of mean sea level pressure over the sub-continent and adjacent south
Atlantic ocean for A) summer (10-13 December 2013) and B) winter (6-10 July 2013)
illustrating the synoptic circulation that results in the transport of air masses from sector G5-
G7.
A

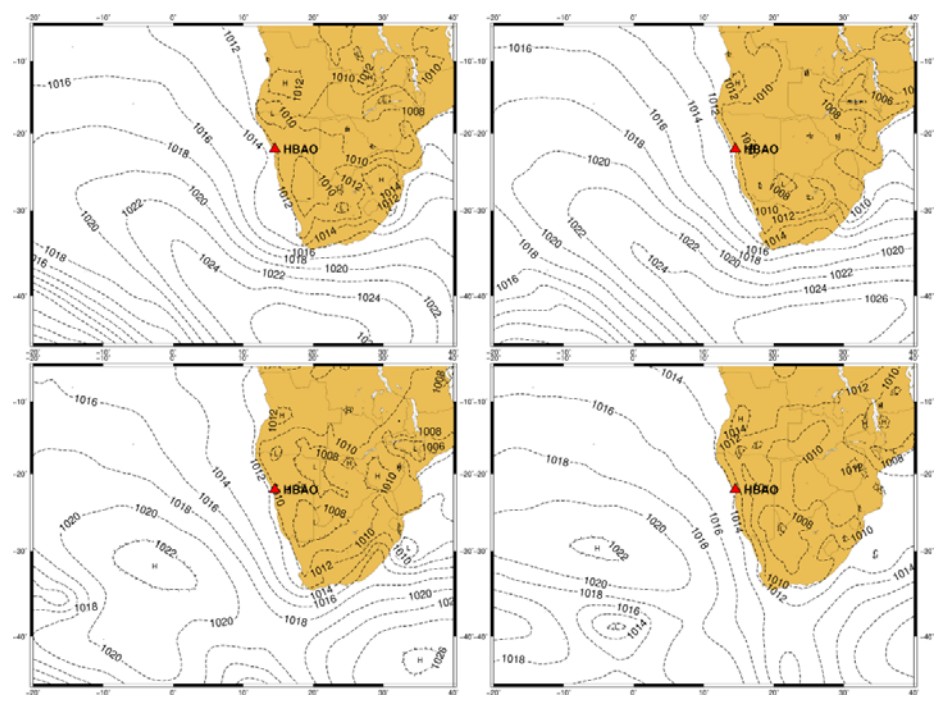


B

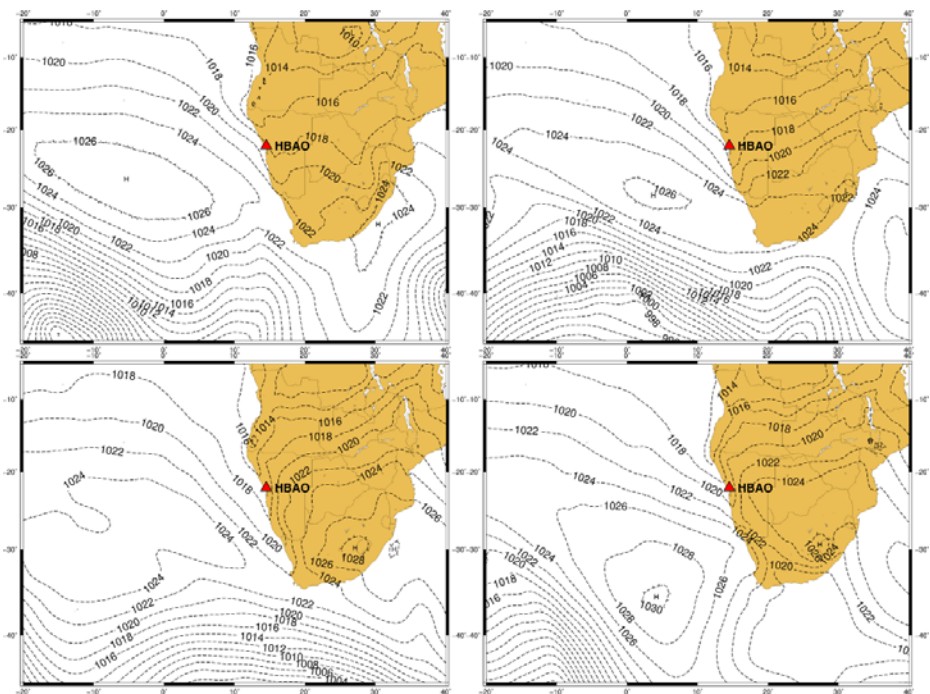


**Figure 9**. Case study of mean sea level pressure over the sub-continent and adjacent South
Atlantic Ocean for 13-16 December 2012 illustrating the synoptic circulation that results in the
transport of air masses from sector G8.

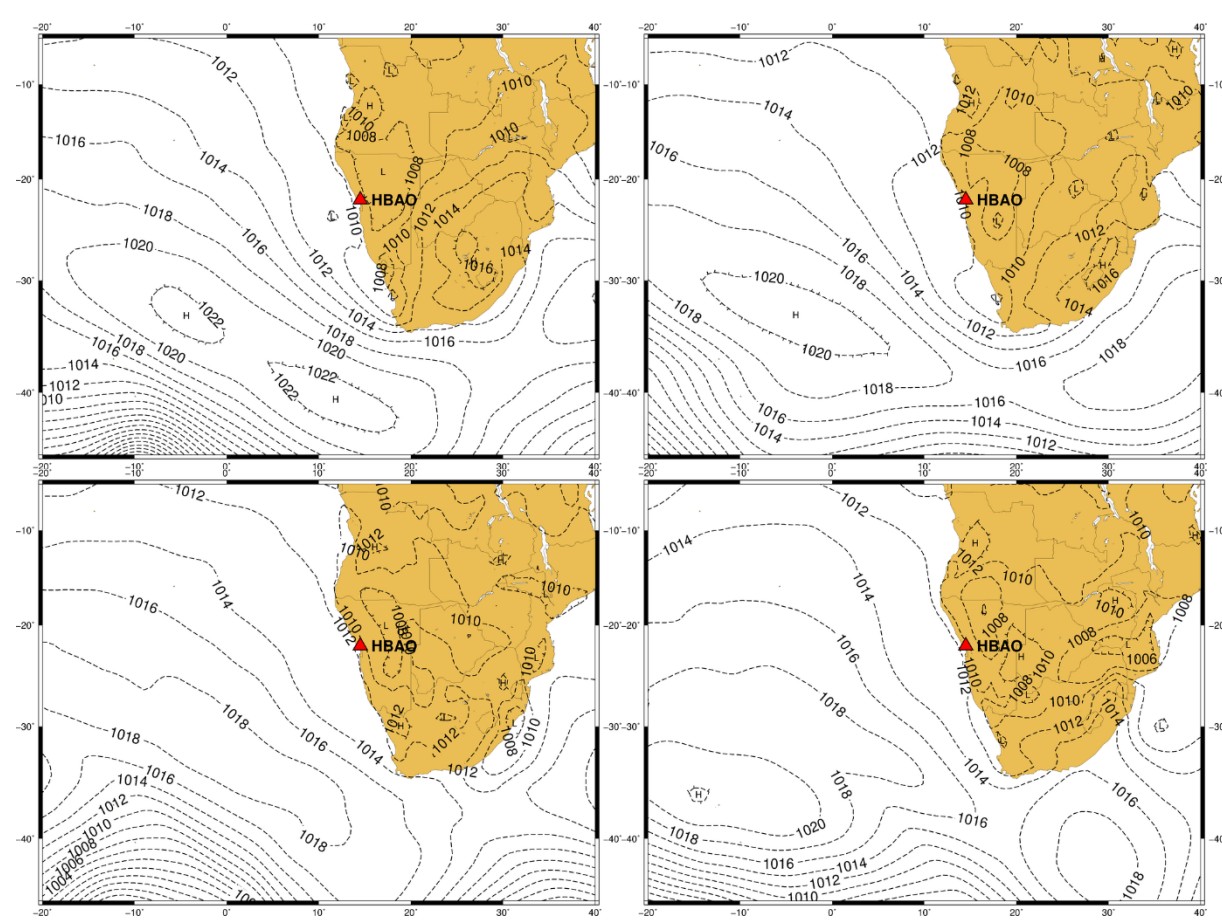



**Figure 10**. Contribution of air mass sectors to the eBC concentrations at HBAO from (a) our
measurements and (b) MERRA-2 reanalysis.

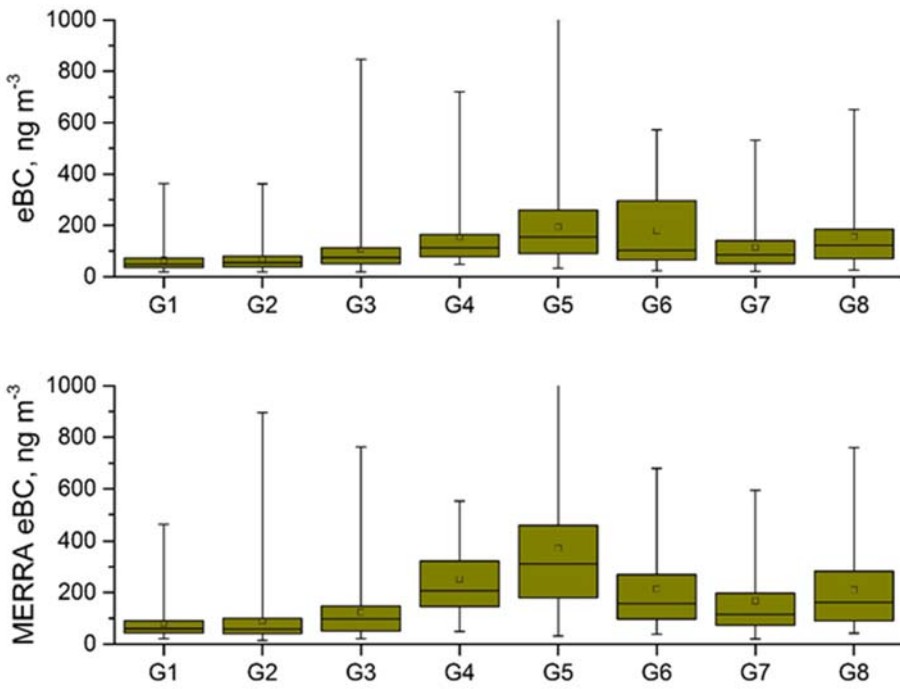
