# Peer review of "Three years of measurements of light-absorbing aerosols over coastal Namibia: seasonality, origin, and transport"

_Atmospheric Chemistry and Physics, 2017_

## Referee Comment (RC1) · Anonymous Referee #1 · 23 Jun 2017

Review of Formenti et al., Three years of measurements of light-absorbing aerosols in the marine air at Henties Bay, Namibia: seasonality, origin, and transport, Atmos. Chem. Phys. Discuss., https://doi.org/10.5194/acp-2017-471, 2017.

This paper describes long-term eBC measurements at a coastal site in Namibia. The authors put the measurements in the context of atmospheric flow using back trajectories and meteorological maps. What I really like about this paper is that it takes the next step – it goes from observations (which are important, but not necessarily exciting on their own) to trying to assess the implications of the observations for radiative forcing using some rough calculations. The calculations are rough due to the limited

nature of the current data set, but suggest this could be an interesting site to continue measurements with a larger suite of instruments.

Major comments: Line 121-122: "our best-guess estimate for sigma_BC in this work is 4.6 ($\pm$ 0.8) m2 g-1 by extrapolating at 880 nm the mean value of Bond et al. (2013)." I think a little more reasoning of why this is your best guess would be appropriate. The Bond value is so much lower than the other values you cite for biomass burning in the region and in the Amazon. Are you not dealing with biomass aerosol or is it aged and those numbers were for fresh or were there potential limitations in the studies cited (Liousse, Kirchsteter and Martins studies)?

Lines 163-165: did you look at whether the trajectories change during biomass burning season (Aug-Oct) relative to the peak eBC time period (May-Aug)? The manuscript discusses winter and summer differences (although none of the figures show the winter summer differences), but might be interesting to look at side-by-side seasons? And or how the trajectory clusters change as a function of season. Which trajectories are most common in the different seasons? Perhaps you could make a table showing total number of trajectories studied and then how many of each type fit into each season?

Line 278 Equation 2 doesn't appear to account for hygroscopicity which seems like it would be important at a coastal site (fm=10% seems like a good estimate of absorption to scattering for dry aerosol, corresponding to an SSA of 0.9)

Lines 276-292: Is the AOD of 0.01 comparable to the AOD measured by collocated AERONET measurements during clean periods? Does the discrepancy between the estimate of AOD from the eBC measurements and AOD from AERONET suggest something about aerosol aloft (above the BL) or do you think it's an artifact of not accounting for hygroscopicity (or some combination of the two). I realize this is meant to be a simplistic calculation, but if the results are different by a factor of 20 (0.01 vs 0.2) some comment should made evaluating the comparison. Perhaps an alternate approach would be to figure out what eBC concentration would correspond to the observed AOD values from AERONET and whether eBC concentrations of that size are seen.

Line 311: Andreae et al (1995) suggest a range of 2-14 cm-3 (ng C m-3)-1 for 'b'. Also Andreae et al used a mass absorption cross section of 10 m2/g for their aethalometer measurements (presumably also at 880 nm for aeths of that time period, although it could've been a broad spectrum instrument). Again, I realize this is a simplistic and speculative calculation (and I think it's great you are doing it!) but a little more description of why you chose 14 rather than 10 or 5 for 'b' would be good as well as some consideration of the potential differences between the eBC from your measurements and Andreae's measurements would be good (since you have that information). Perhaps could move lines 334-341from conclusions into this discussion.

Editorial and minor comments:

Line 68: "..aerosol particles are conducted.." change to "..aerosol particles have been conducted..."

Line 71: "The research centre is located on the.." change to "The research centre is located in (or within) the..." (I'm not sure if in or within is better.)

Line 75: River mouth doesn't need to be capitalized.

Line 81: give diameter and flow rate through inlet pipes?

Line 87: "...through the laden filter..." change to "through the particle laden filter"

Line 100: Add a sentence between the first and second sentence: "This assumption is supported by measurements of other potentially absorbing aerosol in the region.

Line 100: delete "As a matter of fact,"

Line 105: "..fraction should systematically.." change to "..fraction would need to systematically.."

Line 105-106: cause equivalent absorption to what? How many ug/m3 of black carbon?

Line 121-122: "...4.6 (± 0.8) m2 g-1 by extrapolating at 880..." change to "...4.6 (± 0.8) m2 g-1 obtained by extrapolating at 880..."

Line 135: "in the following SAWS, 2016..." change to "...hereafter referred to as SAWS, 2016..."

Line 143: We define "..excess eBC mass concentrations" the occurrences.." change to ""..excess eBC mass concentrations" as the occurrences.."

Line 149-150: these sentences are a little confusing – is the agreement remarkable because the amount of eBC was similar (50-150 ug/m3)? Since Andreae's sigma_BC was ~2x larger than the sigma_BC assumed for your data set the agreement seems potentially coincidental rather than remarkable. Or did you mean that it's remarkable that this manuscript and Andreae's paper both indicated a strong continental influence in pristine locations?

Line 155: delete "On the other hand,"

Line 162: "..May to September.." later you say "..May to August.."

Line 182: define the winter and summer months - is winter JJA and summer DFJ?

Line 199 and 204: I don't think West Coast needs to be capitalized

Line 202: I don't think Westerly Wave needs to be capitalized

Line 212-214: "..the seasonal difference of the boundary layer height, also associated with the changes in synoptic regimes, is opposite to the seasonal cycle of the BC concentrations. This is confusing. Figure 2 shows the highest concentrations of eBC in austral winter JJA and lowest in austral summer. Lines 211-212 says the BL is 1000 m in winter and and 500 m in summer. That suggests the BL is high when the eBC is high and vice versa and thus has the same seasonal cycle as eBC.

Line 219: say how many trajectories this ended up being.

Lines 269-275: Seems like you could also look at the AAOD using an equation similar to equation 2 without the fm factor and using the assumed value for sigma_BC. For AAOD you could ignore hygroscopicity (well, it's commonly done to ignore hygroscopicity of absorbing aerosol – whether it's a good idea to do so is another question!).

Line 287: you used 700 m as delta z – but earlier you discuss boundary layer heights of 500 m in summer and 1000 m in winter. Why not use one of those values (or both) to get a range?

Line 292: start a new paragraph.

Line 292: "However, that the long-range…" rewrite "While estimates of AOD from eBC suggest little radiative effect, the long-range…"

Line 294: "…consistent to measurements…" change to "…consistent with measurements…"

Line 315: "..therefore exceeding by.." change to "…exceed by…"

Line 332: "Indeed, as the specific attenuation used in calculating the concentrations of black carbon from optical attenuation measurements are different, comparisons have to be considered as indicative." Change to "However, because the specific attenuation used in calculating the concentrations of black carbon from optical attenuation measurements are different in Andreae et al (1995) and the current study, such comparisons have to be considered very rough.

Lines 345-353 should go in the main body of the manuscript. (Much of the time I was reading I was wondering what the aerosol particles were if they weren't biomass burning during the dry season). Probably the best place for these lines is a new paragraph at the end of section 3.1. You might have to add some text referring to the next section about circulation, something along the lines of: "The existence of two major transport patterns (anticyclonic recirculation and along-the-coast streamlines) as described in

section 3.2 below suggests. . ."

Lines 355-356: "Their direct radiative effect should be insignificant." Rewrite to "This low AOD value suggests their radiative impact should be insignificant."

Lines 358-363: I would move these lines to line 295 after the (Junkermann and Hacker, 2015) reference. Then I would start a new paragraph with the sentence 'This evaluation. . ."

Line 363: I would start a new bullet point with "Using an empirical relationship. . ."

Figure 2: I think it would be useful to have a second plot or an inset plot that shows the overall seasonality. See for example, figure 6c in http://aaqr.org/files/article/320/31_AAQR-15-05-SIMtS-0358_855-872.pdf which shows both the time series and the seasonality of eBC.

Figure 2 – could you change the tickmarks so there are 12 sections in each year? Right now there are 11 so it's hard to figure out months.

Figure 5 and figure 4.S could be combined – you could put the box whisker from figure 5 (oriented horizontally) above the trajectory distribution in figure 4.S. You would need to use the same x-axis range for MBC. Note: should label eBC rather than MBC for consistency with rest of paper. The paragraph (lines 250-263) discusses figure 4.S enough that I think it should be included in the main text.

---

## Referee Comment (RC2) · Anonymous Referee #2 · 9 Aug 2017

**Review of "Three years of measurements of light-absorbing aerosols in the marine air at Henties Bay, Namibia: seasonality, origin, and transport" by Formenti et al.**

The authors present the results of three years of aethalometer measurements at a coastal site in Namibia. They provide a statistical analysis and relate the observed concentrations to airmass trajectories and synoptic meteorology. From their measurements of aerosol light attenuation, they derive estimates of black carbon concentration, aerosol optical depth, and cloud droplet number concentrations, and consider their effects on the climate system.

Given the scarcity of measurements in this remote region, this is a valuable dataset, which deserves publication. I feel, however, that some of the interpretations stretch the reliability of the estimates beyond their limits and should either be removed or provided with a lot more caveats, based on quantitative assessment of the uncertainties. I therefore recommend a major revision of the manuscript. Specific comments follow:

1) The paper is based on aethalometer measurements, which are converted to equivalent black carbon (eBC) concentrations. The conversion of the measured attenuation (ATN) to eBC is a highly uncertain process, which has been the subject of numerous studies and considerable controversy. In their discussion of the attenuation cross section (l. 107 – 123), the authors mix attenuation and absorption cross sections, which are not the same thing.

2) For the actual conversion of ATN to eBC, the authors use the correction of Weingartner et al. (2003). This is the oldest and probably least accurate formulation. There are numerous later papers (e.g., Virkkula et al., 2007; Schmid et al., 2006; Arnott et al., 2005; Collaud Coen et al., 2010; Saturno et al., 2017), which improve on the Weingartner approach. Especially Collaud Coen et al. (2010) and Saturno et al. (2017) provide evaluations of the earlier approaches. The authors should justify the choice of their correction techniques and provide a quantitative assessment of the resulting errors. They should also avoid given numbers to inappropriate precision, e.g., the value of 986 ng m$^{-3}$, which implies an accuracy of three digits but is probably uncertain to a factor of two.

3) The authors present a detailed analysis of airmass transport patterns. The value of this discussion would be considerably enhanced if these patterns would be related to source distributions. For example, show maps of emissions from biomass burning and power productions. Several gridded inventories are available, which could be used for this purpose.

4) In section 4, the authors calculate a mean AOD value from their eBC concentration estimates. For this purpose, they use values of the mass fraction of eBC to total aerosol mass (PM2.5 ?), aerosol mass extinction efficiency, and boundary layer thickness. All of these parameters vary by factors of 50% to 200%. The authors must provide uncertainty estimates of the individual parameters and use quantitative error propagation to provide an uncertainty estimate of their AOD estimate.

5) The estimated AOD due to the eBC associated aerosol is only 0.01, while the measured AOD at the site is 0.2 to 0.4. This requires most of the aerosol to be outside of the boundary layer. That is of course possible in principle, but should be justified. What data are there in the literature that show elevated aerosol layers of this AOD in the study region?

6)  The biggest stretch comes when the eBC data are used to estimate cloud droplet concentrations (CDNC). Here, two parameters are used, the CN to BC ratio from Andreae et al. (1995) and the CNDC/AMNC ratio of Hegg et al. (2012). The former number was obtained with fairly ancient instrumentation and the paper actually states *"the ratio of CN to black carbon varies between 2 and 14 $cm^{-3}$ (ng C $m^{-3}$)$^{-1}$. The highest ratio was found in one episode on 18 March, close to the African continent during the more remote episodes the ratios were well below 10 $cm^{-3}$ (ng C $m^{-3}$)$^{-1}$. Measurements over Africa during the dry season also showed ratios of about 2-4 $cm^{-3}$ (ng C $m^{-3}$)$^{-1}$ …"*. In other words, the uncertainty of this parameter is about a factor of seven at face value, given the accuracy of the measurements at the time, probably worse. Note that this parameter also refers to CN, not to "AMNC", the accumulation mode number concentration, which is the variable used by Hegg et al. AMNC is typically much smaller than CN, by an unknown and variable factor. So, AMNC could well have been 200, instead of the 2900 $cm^{-3}$ inferred by the authors. Furthermore, assuming the AMNC were as high as the authors estimate, the would be well above the range considered by Hegg et al., and in the range where the CDNC has already saturated because of water vapor limitations. It is simply not legitimate to extrapolate this linear relationship into a range where it does not apply. In conclusion, the uncertainty of this estimate is so large that it probably exceeds one order of magnitude, and this section really should be removed.

---

## Referee Comment (RC3) · Anonymous Referee #3 · 10 Aug 2017

The paper presents results from a 3-year record of atmospheric Black Carbon measured at a coastal site in Namibia using an aethalometer instrument. The record is presented together with a variability analysis and connected to synoptic meteorology in the region and air-mass back trajectories. Further analysis of the results is used to derive BC contribution to aerosol optical depth and potential contribution of BC to cloud droplet number concentrations as an estimate of BC role in regional climate forcing. While it is clear that data documenting BC in areas of the World where information is scarse is very valuable, the paper is, to my view, not ready for publication : on one side, analysis of the observed variability is of limited scope and, in the other, the atmospheric relevance section goes way to far beyond reasonable interpretation of the

actual measurements. Section 4. is not acceptable as it is : going from ATN measurements to Cloud Number Concentration using fixed proportionality factors derived from 2 previous studies without any independent way to control the estimate is scientifically questionable. This entire section should be removed unless additional evidences are provided (this applies to third bullet point in the conclusion as well). Other comments : Line 125, P.6 : why using Weingartner et al. when more suited correction procedures are now proposed in litterature ? Line 142, P.6 : on which basis is the 100 ng m-3 treshold chosen as a base for Âń excess eBC mass Âż. This is misleading also when presenting results and figures : for example, it is not clear if 100 ng m-3 should be added to data in Figure 5 to actually get the actual measurements. Similarly, Table 2 (excess) and Table 1 (observed) are not directly comparable. I recommend not to use this 100 ng/m3 treshold consideration and stick to eBC, and not excess eBC. If any Âń excess eBC Âż must be used, it should then be defined upon statistical analysis of the record as definition of background conditions is not so trivial. Line 160, P.7 : seasonal variability is more than just apparent in Figure 2. In fact, adding a Figure/Table with the actual monthly values would help (in fact, documenting diurnal variability would also help). Section 3.2.2 : I have difficulties connecting section 3.2.1 to section 3.2.2. If seasonal dependent synoptic circulation controls BC variability, this should also appear somehow in the back-trajectory analysis, which is not mentioned. I am surprised not to see any artefact in Figure 4 due to the lack of measurements during the first 6-month in 2015 (not explained). In this section, linking concentration values for each flow pathways to potential sources identified through relevant emission inventories may add useful information to the study. Table 2 : many other studies can actually be added to the table (also add your own results).

---

## Author Comment (AC1) · 12 Nov 2018

**Answer to reviews for ms acp-2017-471**

Formenti et al., Aerosol optical properties derived from POLDER-3/PARASOL (2005-2013) over the western Mediterranean Sea: I. Quality assessment with AERONET and in situ airborne observations

**https://doi.org/10.5194/acp-2017-471, 2017.**

We thank the three referees for evaluating the manuscript and providing us with feedback on its scientific content. We also apologize for the long delay in addressing these reviews. We have now provided a full revision of the paper.

Referees had common concerns which we addressed to the best of our possibilities:

- We clarified the methodological section discussing the conversion of attenuation to eBC concentrations;

- We simplified and augmented the meteorological section to better explain the links between the measured concentrations and the synoptic scale transport (Figures 5 to 9)

- We added a comparison to the MERRA-2 model reanalysis (Figures 2 and 10)

- We have added emission grid maps and MODIS imagery for fire counts to elucidate the links between measured concentrations and emission sources (Figures 3, 4 and 10)

- We eliminated the "Atmospheric implication" section, simplified the arguments in the "Discussion and Conclusions" section. The calculation of the possible number of could condensation nuclei (CCN) particles was removed

Detailed responses are reported in the body of text here below in blue

**Anonymous Referee #1**

This paper describes long-term eBC measurements at a coastal site in Namibia. The authors put the measurements in the context of atmospheric flow using back trajectories and meteorological maps. What I really like about this paper is that it takes the next step – it goes from observations (which are important, but not necessarily exciting on their own) to trying to assess the implications of the observations for radiative forcing using some rough calculations. The calculations are rough due to the limited nature of the current data set, but suggest this could be an interesting site to continue measurements with a larger suite of instruments.

Major comments: Line 121-122: "our best-guess estimate for sigma\_BC in this work is 4.6 ( $\pm$ 0.8) m2 g-1 by extrapolating at 880 nm the mean value of Bond et al. (2013)." I think a little more reasoning of why this is your best guess would be appropriate. The Bond value is so much lower than the other values you cite for biomass burning in the region and in the Amazon. Are you not dealing with biomass aerosol or is it aged and those numbers were for fresh or were there potential limitations in the studies cited (Liousse, Kirchsteter and Martins studies)?

This section posed problems to all the Reviewers and it has now been reformulated.

Lines 163-165: did you look at whether the trajectories change during biomass burning season (Aug-Oct) relative to the peak eBC time period (May-Aug)? The manuscript discusses winter and summer differences (although none of the figures show the winter summer differences), but might be interesting to look at side-by-side seasons? And or how the trajectory clusters change as a function of season. Which trajectories are most common in the different seasons? Perhaps you could make a table showing total number of trajectories studied and then how many of each type fit into each season?

These sections are now completely reformulated. Sections 3.2 and 3.3 discuss of the general synoptic circulation patterns for the region and link them to the observed air mass pathways for all seasons. Figure 5 present theirs seasonality (monthly variability).

Line 278 Equation 2 doesn't appear to account for hygroscopicity which seems like it would be important at a coastal site (fm=10% seems like a good estimate of absorption to scattering for dry aerosol, corresponding to an SSA of 0.9)

The measurements are conducted at ambient conditions, clearly hygroscopicity could be important at the site.

Lines 276-292: Is the AOD of 0.01 comparable to the AOD measured by collocated AERONET measurements during clean periods? Does the discrepancy between the estimate of AOD from the eBC measurements and AOD from AERONET suggest something about aerosol aloft (above the BL) or do you think it's an artifact of not accounting for hygroscopicity (or some combination of the two). I realize this is meant to be a simplistic calculation, but if the results are different by a factor of 20 (0.01 vs 0.2) some comment should made evaluating the comparison. Perhaps an alternate approach would be to figure out what eBC concentration would correspond to the observed AOD values from AERONET and whether eBC concentrations of that size are seen.

This is actually a very good suggestion by Referee #1, whom we thank. We do not have AERONET measurements during the period of operation of the aethalometer. However, we could calculate the average AOD of the fine mode based on the Spectral Deconvolution Algorithm (SDA) method by O'Neill (https://aeronet.gsfc.nasa.gov/new\_web/PDF/tauf\_tauc\_technical\_memo.pdf).

The mean fine mode AOD at 500 nm for the January to April 2011 is  $0.05 \pm 0.05$ . In the paper, we avoided discussing the conversion to mass concentration, as too many assumptions are involved, and rather discuss what it represents to the total AOD and to the AOD in the main biomass burning season.

Line 311: Andreae et al (1995) suggest a range of 2-14 cm-3 (ng C m-3)-1 for 'b'. Also Andreae et al used a mass absorption cross section of 10 m2/g for their aethalometer measurements (presumably also at 880 nm for aeths of that time period, although it could've been a broad spectrum instrument). Again, I realize this is a simplistic and speculative calculation (and I think it's great you are doing it!) but a little more description of why you chose 14 rather than 10 or 5 for 'b' would be good as well assume consideration of the potential differences between the eBC from your measurements and Andreae's measurements would be good (since you have that information). Perhaps could move lines 334-341 from conclusions into this discussion.

According to recommendations by Reviewers #1 and #3, the calculation of the potential CCN number concentrations have been completely removed.

All editorial and minor comments have been taken into account

Review of "Three years of measurements of light -absorbing aerosols in the marine air at Henties Bay, Namibia: seasonality, origin, and transport " by Formenti et al.

The authors present the results of three years of aethalometer measurements at a coastal site in Namibia. They provide a statistical analysis and relate the observed concentrations to airmass trajectories and synoptic meteorology. From their measurements of aerosol light attenuation, they derive estimates of black carbon concentration, aerosol optical depth, and cloud droplet number concentrations, and consider their effects on the climate system.

Given the scarcity of measurements in this remote region, this is a valuable dataset, which deserves publication. I feel, however, that some of the interpretations stretch the reliability of the estimates beyond their limits and should either be removed or provided with a lot more caveats, based on quantitative assessment of the uncertainties.

I therefore recommend a major revision of the manuscript. Specific comments follow:

1) The paper is based on aethalometer measurements, which are converted to equivalent black carbon (eBC) concentrations. The conversion of the measured attenuation (ATN) to eBC is a highly uncertain process, which has been the subject of numerous studies and considerable controversy. In their discussion of the attenuation cross section (I. 107 - 123), the authors mix attenuation and absorption cross sections, which are not the same thing.

2) For the actual conversion of ATN to eBC, the authors use the correction of Weingartner et al. (2003). This is the oldest and probably least accurate formulation. There are numerous later papers (e.g., Virkkula et al., 2007; Schmid et al., 2006; Arnott et al., 2005; Collaud Coen et al., 2010; Saturno et al., 2017), which improve on the Weingartner approach. Especially Collaud Coen et al. (2010) and Saturno et al. (2017) provide evaluations of the earlier approaches. The authors should justify the choice of their correction techniques and provide a quantitative assessment of the resulting errors. They should also avoid given numbers to inappropriate precision, e.g., the value of 986 ng m-3, which implies an accuracy of three digits but is probably uncertain to a factor of two.

We agree with Reviewer #2 and we have significantly revised Section 2.1 to take caveats into account. However, contrary to Collaud-Coen et al. (2010) and Saturno et al. (2017) we do not have concurrent measurements of the scattering coefficient to constrain the multi-scattering matrix effect. We applied the protocol recommended by Collaud-Coen et al. (2010) for those cases (Figure 1 of their manuscript).

3) The authors present a detailed analysis of airmass transport patterns. The value of this discussion would be considerably enhanced if these patterns would be related to source distributions. For example, show maps of emissions from biomass burning and power productions. Several gridded inventories are available, which could be used for this purpose.

This is now done in Figures 3 and 4 of the manuscript.

4) In section 4, the authors calculate a mean AOD value from their eBC concentration estimates. For this purpose, they use values of the mass fraction of eBC to total aerosol mass (PM2.5 ?), aerosol mass extinction efficiency, and boundary layer thickness. All of these parameters vary by factors of 50% to 200%.

The authors must provide uncertainty estimates of the individual parameters and use quantitative error propagation to provide an uncertainty estimate of their AOD estimate.

5) The estimated AOD due to the eBC associated aerosol is only 0.01, while the measured AOD at the site is 0.2 to 0.4. This requires most of the aerosol to be outside of the boundary layer. That is of course possible in principle, but should be justified. What data are there in the literature that show elevated aerosol layers of this AOD in the study region?

**Figure 2S in the supplementary and the revision of the main text (section "Discussion and conclusions") address these issues.**

6) The biggest stretch comes when the eBC data are used to estimate cloud droplet concentrations (CDNC). Here, two parameters are used, the CN to BC ratio from Andreae et al. (1995) and the CNDC/AMNC ratio of Hegg et al. (2012). The former number was obtained with fairly ancient instrumentation and the paper actually states "the ratio of CN to black carbon varies between 2 and 14 cm-3 ( ng C m-3)-1. The highest r atio was found in one episode on 18 March, close to the African continent during the more remote episodes the ratios were well below 10 cm-3 (ng C m-3)-1. Measurements over Africa during the dry season also showed ratios of about 2-4 cm-3 (ng C m-3)-1...". In other words, the uncertainty of this parameter is about a factor of seven at face value, given the accuracy of the measurements at the time, probably worse. Note that this parameter also refers to CN, not to "AMNC", the accumulation mode number concentration, which is the variable used by Hegg et al. AMNC is typically much smaller than CN, by an unknown and variable factor. So, AMNC could well have been 200, instead of the 2900 cm-3inferred by the authors. Furthermore, assuming the AMNC were as high as the authors estimate, the would be well above the range considered by Hegg et al., and in the range where the CDNC has already saturated because of water vapor limitations. It is simply not legitimate to extrapolate this linear relationship into a range where it does not apply. In conclusion, the uncertainty of this estimate is so large that it probably exceeds one order of magnitude, and this section really should be removed.

This section is now removed from the paper.

**Anonymous Referee #3**

The paper presents results from a 3-year record of atmospheric Black Carbon measured at a coastal site in Namibia using an aethalometer instrument. The record is presented together with a variability analysis and connected to synoptic meteorology in the region and air-mass back trajectories. Further analysis of the results is used to derive BC contribution to aerosol optical depth and potential contribution of BC to cloud droplet number concentrations as an estimate of BC role in regional climate forcing. While it is clear that data documenting BC in areas of the World where information is scarse is very valuable, the paper is, to my view, not ready for publication: on one side, analysis of the observed variability is of limited scope and, in the other, the atmospheric relevance section goes way to far beyond reasonable interpretation of the actual measurements. Section 4. is not acceptable as it is : going from ATN measurements to Cloud Number Concentration using fixed proportionality factors derived from 2 previous studies without any independent way to control the estimate is scientifically questionable. This entire section should be removed unless additional evidences are provided (this applies to third bullet point in the conclusion as well).

We thank Referee #3 for valuable comments which we taken into account to the best of our possibilities. We have removed Section 4 "Atmospheric implications" and in particular the discussion about the CCN-potential of the eBC-laden air masses.

Other comments :

Line 125, P.6 : why using Weingartner et al. when more suited correction procedures are now proposed in litterature ?

This concern was rose by Reviewer #2 too. We have significantly revised Section 2.1 to take caveats into account. However, contrary to Collaud-Coen et al. (2010) and Saturno et al. (2017) we do not have concurrent measurements of the scattering coefficient to constrain the multi-scattering matrix effect. We applied the protocol recommended by Collaud-Coen et al. (2010) for those cases (Figure 1 of their paper).

Line 142, P.6 : on which basis is the 100 ng m-3 threshold chosen as a base for an excess eBC mass. This is misleading also when presenting results and figures : for example, it is not clear if 100 ng m-3 should be added to data in Figure 5 to actually get the actual measurements. Similarly, Table 2 (excess) and Table 1 (observed) are not directly comparable. I recommend not to use this 100 ng/m3 treshold consideration and stick to eBC, and not excess eBC. If any excess eBC must be used, it should then be defined upon statistical analysis of the record as definition of background conditions is not so trivial.

We removed references to excess-concentrations and only present measured values

Line 160, P.7 : seasonal variability is more than just apparent in Figure 2. In fact, adding a Figure/Table with the actual monthly values would help (in fact, documenting diurnal variability would also help).

Figure 2 now also shows the monthly variability

Section 3.2.2 : I have difficulties connecting section 3.2.1 to section 3.2.2. If seasonal dependent synoptic circulation controls BC variability, this should also appear somehow in the back-trajectory analysis, which is not mentioned. I am surprised not to see any artefact in Figure 4 due to the lack of measurements during the first 6-month in 2015 (not explained). In this section, linking concentration values for each flow pathways to potential sources identified

through relevant emission inventories may add useful information to the study. Table 2 : many other studies can actually be added to the table (also add your own results).

These sections are now completely reformulated. We extended the calculations of air mass back trajectories. We simplified the presentation of air mass sectors and refine the presentation of the synoptic meteorology controlling them. We present their seasonality (monthly variability) and evaluate the seasonality of eBC contribution by sectors. We also added a comparison to MERRA-2 reanalysis.

---

## Author Response (AR2)

**Answer to co-editor for ms acp-2017-471**

Dear Andreas

We have corrected the typographical error and ameliorated figures as requested.

In doing so, we also made three additional modifications

1/ corrected the figure numbering that was inconsistent between the Figure caption list and the figures themselves

2/ modified the legend of Figure 2 as follows (in red the added words): "Figure 2. Top panel (a): comparisons of the time series of daily eBC mass concentrations (ng m-3) measured at HBAO and predicted by the MERRA-2 reanalysis. The light grey boxes indicate periods of increasing concentrations. The light blue boxes indicate periods of decreasing concentrations. Bottom panel (b) box and whisker plot representation of the respective monthly variability."

3/ modified the axis of figure 5 which by mistake where not in %

Thank you so much for the prompt acceptance of the manuscript.

Best regards.

Paola